# Maternal cytomegalovirus infection and delayed language development in children at 3 years of age–a nested case-control study in a large population-based pregnancy cohort

Regine Barlinn[1][ID][☯][*], Susanne G. Dudman[2][ID][☯], Halvor Rollag[2], Lill Trogstad[3], Jonas C. Lindstrøm[4], Per Magnus[5,6]

**1** Division for Infection Control and Environmental Health, Department of Microbiology, Oslo University Hospital, Norwegian Institute of Public Health, Oslo, Norway, **2** Department of Microbiology, Oslo University Hospital, and University of Oslo, Oslo, Norway, **3** Division for Infection Control and Environmental Health, Norwegian Institute of Public Health, Oslo, Norway, **4** Division for Infection Control and Environmental Health, Department of Methods Development and Analytics, Norwegian Institute of Public Health, Oslo, Norway, **5** Centre for Fertility and Health, Norwegian Institute of Public Health, Oslo, Norway, **6** University of Oslo, Oslo, Norway

☯ These authors contributed equally to this work.
* regbar@ous-hf.no

## Abstract

### Introduction

Maternal cytomegalovirus (CMV) infection in pregnancy may result in vertical transmission of CMV to the child. Long-term effects of congenital CMV infection include visual, cognitive as well as neurological impairment. The aim of this study was to estimate the odds ratios for CMV seropositivity and seroconversion in mothers, with and without delayed language development in 3 year old children, nested within a large cohort.

### Material and methods

The Norwegian Mother, Father and Child Cohort Study (MoBa) is a prospective population-based pregnancy cohort that includes 95 200 mothers and 114 500 children. Blood samples were obtained from mothers during pregnancy weeks 17 or 18 in pregnancy and after birth. We included 300 women from MoBa with children suffering from delayed language development at three years of age, based on validated questionnaires. Within the cohort, 1350 randomly selected women were included as controls to perform a nested case-control study. The cases and controls were tested for CMV IgG antibodies by an enzyme-linked immunosorbent assay.

### Results

Among mothers of cases, 63.2% were CMV-IgG positive in the sample at birth, as compared to 55.9% among controls; OR 1.36, (95% CI; 1.05 to 1.76). Also, among case mothers, 8/118 (6.8%) initially seronegative cases, seroconverted. Among initially seronegative controls, seroconversion occurred in 23/618 (3.7%) mothers. The OR for seroconversion in

**Data Availability Statement:** Data from this study are available only upon request as there are legal and ethical restrictions on sharing data publicly. Data cannot be made publicly available because it contains sensitive participant information. Additionally, participants did not give consent for data to be made publicly available in a repository. Interested researchers can request access to the relevant datasets via request to: datatilgang@fhi. no. The request will require approval from an ethics committee and formal contract with Norwegian Mother and Child Cohort study (MoBa).

**Funding:** Funding The Norwegian Mother and Child Cohort Study is supported by the Norwegian Ministry of Health and Care Services and the Ministry of Education and Research, NIH/NINDS (grant no.1 UO1 NS 047537-01 and grant no.2 UO1 NS 047537-06A1). This study was funded by a grant (213916/H10) from the Norwegian Research Council.

**Competing interests:** The authors have declared that no competing interests exist.

cases as compared to control mothers was 1.88 (CI; 0.82 to 4.31), thus not statistically significant different.

## Conclusion

This study shows a higher risk of delayed language development at three years of age in children born by mothers seropositive for CMV, compared to children born from seronegative mothers.

## Introduction

Maternal cytomegalovirus (CMV) infection in pregnancy may result in vertical transmission of CMV to the fetus [1]. CMV is the most common congenital infection and affects 0.7% of all newborns [2]. The virus may be vertically transmitted both in primary and non-primary (reinfection and reactivation) CMV-infections. The transmission rate is between 30–40% during primary infection, which is high compared to around 1% in a non-primary infection [2, 3]. Despite the lower transmission rate, non-primary infections contribute to the majority of congenital infections, given that the seroprevalence is above 30% [1]. The CMV seroprevalence varies in different parts of the world and is influenced by age and socioeconomic factors [4]. In Norway, the CMV-IgG seroprevalence in pregnant women lies around 60% with a seroconversion rate of 3.7% [5, 6]. Of all infected newborns, 15–20% suffer from long-term sequelae. CMV is a leading cause of birth defects and developmental disabilities [7]. Congenital CMV infection (cCMV) is the most common cause of nonhereditary sensorineural hearing loss in childhood. Around 12% of cCMV cases will experience hearing loss [8]. Long-term impairment after cCMV may also include visual, cognitive as well as other neurological disabilities [7]. A Dutch study showed that speech and language problems were common in both children with cCMV and children without cCMV. Nevertheless, children with cCMV were most frequently affected [9]. Studies with long follow up are needed to gain knowledge on outcomes developing over several years in newborns with cCMV.

The aim of this nested case-control study was to estimate the association between CMV seropositivity and seroconversion in pregnant women with and without children who had delayed language development at the age of 3 years, within a large, population-based pregnancy cohort.

## Materials and methods

### Study population

The Norwegian Mother, Father and Child Cohort Study (MoBa) is a prospective population-based pregnancy cohort conducted by the Norwegian Institute of Public Health (NIPH). The main aim of MoBa is to provide a comprehensive database for understanding the causes of serious diseases among Norwegian women and children. Participants were recruited from all over Norway from 1999–2008 after giving informed consent [10, 11]. The cohort includes 95 200 mothers, 75 000 fathers and 114 500 children. Blood (plasma) samples were obtained from mothers in week 17 or 18 during pregnancy (M1) when inclusion was done and after birth (M2) [12]. The data used in this nested case-control study is based on Medical Birth Registry of Norway (MBRN) records and three questionnaires (Q) that the women responded to during pregnancy, and one when the child was 3 years old.

**Table 1. In question 18 of the questionnaire at 3 years of age, mothers are asked which of the six response categories best described the speech performance of their child.**

| Response categories in question 18 |
| --- |
| **1**- Not yet talking |
| **2**- He/she is talking, but you can't understand him/her |
| **3**- Talking in one-word utterances, such as "milk" or "down" |
| **4**- Talking in 2- to 3-word phrases, such as "me got ball" or "give doll" |
| **5**- Talking in complete sentences, such as "I got a doll" or "can I go outside?" |
| **6**- Talking in long and complicated sentences, such as "when I went to the park, I went on the swings" or "I saw a man standing on the corner". |

## Case definition

We included 300 women who had children suffering from delayed language development at three years of age. The case definition was defined as severe (tick on point 1–3) or moderate (tick on point 4) problems in question 18, and a simultaneous score of ≤40 in question 21 in the 3 year questionnaire [13]. One case mother later withdrew from the study and was excluded, leaving 299 cases eligible for analysis (Tables 1 and 2).

## Controls

Within the MoBa cohort, 1350 randomly selected women, with the birth registered in the MBRN were included as controls. One control mother later withdrew from the study and was excluded, leaving 1349 controls eligible for analysis. After starting to analyse, it became evident that one control had only the M2 sample.

## Further inclusion and exclusion criteria for cases and controls

All included women had given birth to a live born child, had available plasma samples and had completed questionnaire 1 (completed at week 17) and questionnaire 6 (completed at 3 years of age). We excluded children from multiple pregnancies, children with chromosomal abnormalities as registered in the MBRN, and children selected to participate as cases or controls in another ongoing study within MoBa, the autism birth cohort (ABC) study [14].

**Table 2. In question 21, in the questionnaire at 3 years of age, mothers are asked to answer the six questions listed under in either "yes, often", "sometimes", or "not yet", according to whether the child can do the activity ("yes" scored 10 points, "sometimes" = 5 points and "not yet" scored 0 points).**

| Question 21 |
| --- |
| **1**.Without showing him/her first, does your child point to the correct picture when you say, "Where is the cat" or "Where is the dog"? |
| **2**.When you ask your child to point to his/her eyes, nose, hair, feet, ears, and so forth, does he/she correctly point to at least seven body parts? |
| **3**.Does your child make sentences that are three or four words long? |
| **4**.Without giving him/her help by pointing or using gestures, ask your child to "Put the shoe on the table" and "Put the book under the chair". Does your child carry out both of these directions correctly? |
| **5**.When looking at a picture book, does your child tell you what is happening or what action is taking place in the picture? (For example, "Barking", "Running", "Eating" and "Crying"?) You may ask, "What is the dog (or boy) doing?" |
| **6**.Can your child tell you at least two things about an object he/she is familiar with? |

## Outcome measures

The outcome measure based on the maternal response to the questionnaire at 3 years of age. The questions were derived from the Ages and Stages Questionnaire (ASQ) communication scale, which is found to show good test-retest agreement and concurrent validity [15, 16].

In addition we used a language grammar rating scale, in which the mother was asked to choose 1 of 6 categories, ranging from no word production to full sentences with complete grammatical markings [17]. A validation was conducted to assess the questionnaire's ability to detect language development by comparing the parent's response to concomitant psychological examination of the child at 3 years showing high correlation [18]. The six categories of question 21 were derived from the Ages and Stages Questionnaires and were designed to assess the child's ability to comprehend and convey.

## Other variables

The impaired hearing variable was based on the questionnaire at 3 years of age with the question; has your child ever suffered or is currently suffering from impaired hearing.

Maternal educational attainment was scored as a dichotomous variable as completed / ongoing education level ≤12 years or completed / ongoing education level ≥13 years.

## Serological tests

Samples collected from the pregnant women at birth (M2 sample) were analysed for CMV IgM and IgG antibodies by an enzyme-linked immunosorbent assay (Institut Virion/Serion Gmbh, Wuerzburg, Germany). No further testing was done if the M2 sample had an IgG and IgM negative result. The M1 sample was analysed for IgG and IgM in the remaining mothers. Further laboratory details have been described previously [6]. CMV seropositivity was defined as the proportion of women who had detectable IgG antibodies in the sample at birth. Serology results with an equivocal IgG result in M2 samples were interpreted as seronegative (applicable for 1 case and 2 controls). Seroconversion was defined as CMV-IgG antibody negative result in the sample taken in week 17–18 of pregnancy combined with a CMV-IgG antibody positive result in the sample taken at birth.

## Statistical analyses

Logistic regression analysis was used to estimate the effect of CMV IgG seropositivity and seroconversion on delayed language development. Odds ratios (ORs) with accompanying 95% confidence intervals were calculated with univariate logistic regression models, and adjusted ORs were calculated by multivariate logistic regression adjusting for hearing impairment, age, parity, maternal education level, and hearing impairment in the child. Student's t-test was used to calculate p-values for continuous variables. Separate analyses without adjustment for hearing impairment was done because it is not clear if it is a mediator between CMV and language development. In addition, sensitivity analyses where children with hearing impairment were excluded was also done. The sample size needed to perform this study was calculated based on the CMV seroprevalence. A sample of 300 cases and 1350 controls was needed to detect an odds ratio (OR) for CMV-infection of 1.5 with a significance level of 0.05 and power of 80% (OpenEpi.com). Statistical analyses were conducted using SPSS for windows, 27, SPSS Inc. Chicago, IL.

## Details of ethical approval

Informed consent was obtained from each MoBa participant upon recruitment. The establishment and data collection in MoBa obtained a license from the Norwegian Data Inspectorate

(01/4325) and approval from The Regional Committee for Medical Research Ethics (S-97045, S-95113). The current study was approved by the Regional Committee for Medical Research Ethics in South-Eastern Norway (2012/374B).

## Results

Maternal age was similar for case mothers and control mothers, and for seropositive and sero-negative women. The mean gestational length was comparable in the two groups. The maternal education level was lower in case mothers (49%) ≤12 years than in controls (27%). The seropositivity rate was higher in the mothers with lower education level. Fewer case mothers (36.5%) were primiparous compared to controls (48%) and primiparous women were more often seronegative. Mean BMI were higher in cases versus controls, 24.7 and 23.7 respectively. Among cases 43.5% had BMI > 25, compared to 30.9% in controls. In all analyses, parity was positively associated with delayed language development while mother's education level was negatively associated. Hearing impairment in children was strongly associated (OR = 4,32, CI 2.79–6.75, p<0.001) with delayed language development. In children of the case group 13.5% had impaired hearing compared to only 3.5% in children of control mothers. Further characteristics are reported in Table 3.

Among the case mothers, 63.2% were CMV-IgG positive in the sample at birth, as compared to 55.9% among controls, *OR 1.36, (CI; 1.05 to 1.76)* (Tables 4 and 5, model 1). A similar, but not statistical significant estimate was found when adjusted for maternal age, parity, and education level (OR = 1.29, 95% CI 0.99–1.69, p = 0.06) (Table 5, model 2). Additional adjustment for hearing impairment also gave similar results (Table 5, model 3).

Since hearing impairment can be either a confounder or a mediator for delayed language development, we performed the analysis both with and without adjusting for child's hearing impairment giving very similar results (Table 5). A separate analysis of hearing impairment as the outcome, showed a non-significant association between CMV-IgG seropositivity, both in adjusted (OR = 1.08, CI: 0.69–1.70, p = 0.73) and unadjusted (OR = 1.05, CI: 0.68–1.61, p = 0.83) analyses.

In addition, we performed a sensitivity analysis, and excluded mothers of cases and controls with hearing impairment. Among mothers of cases without hearing impairment, 63.7% were CMV- IgG positive, as compared to 55.7% among controls without hearing impairment, still showing a significant positive association between CMV-IgG positivity and delayed language development (OR = 1.40, CI: 1.06–1.85, p = 0.02).

Among case mothers, 8/118 (6.8%) initially seronegative mothers seroconverted. Among initially seronegative control mothers, seroconversion occurred in 23/618 (3.7%), *(OR = 1.88, 95% CI 0.82 to 4.31, p = 0.135)* (Tables 4 and 6, model 1).

When adjusted for maternal age, parity, and education level variables, the association between seroconversion during pregnancy and delayed language development was quite similar to the unadjusted analyses (OR = 2.18, 95% CI: 0.92–5.17, p = 0.08) (Table 6, model 2). Additional adjustment for hearing impairment also gave similar results (Table 6, model 3).

## Discussion

In this nested case-control study from a large, population-based pregnancy cohort, we observed an association between development of language impairment in children at three years of age and CMV IgG positive status of the mother during pregnancy, also present when adjusted for maternal age, parity, and education level. We found a higher seroconversion rate of 6.8% in initially seronegative case mothers compared to controls (3.7%), although the difference was not statistically significant.

**Table 3. Maternal and pregnancy characteristics in 299 delayed language delay cases and 1349 population based controls from the Norwegian Mother, Father and Child Cohort Study (MoBa).**

| Maternal and pregnancy characteristics | | Cases | Controls | P-values[3] | CMV-IgG | CMV-IgG |
|---|---|---|---|---|---|---|
| | | n = 299 | n = 1349 | | seropositive | seronegative |
| | | (%) [SD] | (%) [SD] | | n = 943 (%) | n = 705 (%) |
| Mean age(years)[1] | | 30.9[4.6] | 30.4[4.4] | 0.14 | 30.6 | 30.4 |
| Primiparous (%)[1] | Yes | 109 (36.5) | 647 (48.0) | <0.001 | 400 (42.4) | 356 (50.5) |
| | No | 190 (63.5) | 702 (52.0) | | 543 (57.6) | 349 (49.5) |
| Folic acid intake during pregnancy[2] | Yes | 165 (55.2) | 806 (59.7) | 0.17 | 545 (57.8) | 426 (60.4) |
| | No | 134 (44.8) | 543 (40.3) | | 398 (42.2) | 279 (39.6) |
| Maternal education level*[1] | ≤12 years | 147 (49.2) | 372 (27.6) | <0.001 | 322 (34.1) | 197 (27.9) |
| | ≥13 years | 146 (48.8) | 957 (70.9) | | 603 (63.9) | 500 (70.9) |
| Missing | | 6 | 20 | | 8 | 18 |
| BMI** | | | | | | |
| Mean (range) [1] | | 24.7[6.5] | 23.7[5.4] | 0.002 | 23.8 | 24.0 |
| BMI >25 | Yes | 127 (43.5) | 409 (30.9) | <0.001 | 307 (32.6) | 229 (32.5) |
| | No | 165 (56.5) | 913 (69.1) | | 612 (65) | 466 (66.1) |
| Missing | | 7 | 27 | | 24 | 10 |
| Gestational length, (mean) weeks, (min-max) [2] | | 39.4(30–43) [1.9] | 39.7 (31–44) [1.6] | 0.014 | 39.6 (31–44) | 39.7 (30–43) |
| Birth weight (mean)kg[2] | | 3.567[612.0] | 3.628[506.1] | 0.07 | 3.613 | 3.635 |
| Missing | | 15 | 53 | | 39 | 29 |
| Placenta weight (mean)kg[2] | | 0.668[150.6] | 0.683[194.4] | 0.212 | 0.681 | 0.680 |
| Missing | | 15 | 53 | | 39 | 29 |
| Impaired hearing***[1] | Yes | 39 (13.5) | 47 (3.5) | <0.001 | 50 (5.4) | 36 (5.2) |
| | No | 249 | 1292 | | 878 | 663 |
| Missing | | 11 | 10 | | 15 | 6 |

n = number, SD = standard deviation.

[1] Data from questionnaires.

[2] Data from MBRN.

[3] p-values comparing cases and controls.

*Completed or ongoing education level.

** Pre-pregnancy body mass index, (BMI).

***Data collection is based on the questionnaire when the child was 3 years of age; has your child ever suffered, or is currently suffering from hearing problems.

Speech and language problems are common in children and have multiple causes, the most severe are caused by chromosomal anomalies, inherited disorders and injuries at birth. Hearing loss, which is a common outcome in children with cCMV, can lead to speech and language impairment. However, studies performed in both asymptomatic and symptomatic subjects have shown that impairment in language development may also be present in cCMV children without hearing loss [19, 20]. This support that different parts of the brain has been affected in children with delayed language development compared to those with hearing loss [21]. That delayed language development in cCMV can occur in children without hearing loss is also described in the DECIBEL study by Korver and coworkers [22]. They found a greater delay in language comprehension in children with both permanent childhood hearing impairment and cCMV, than in children with permanent childhood hearing loss without cCMV. Another recent study also showed that both children with an asymptomatic or a symptomatic cCMV infection developed long-term sequelae, particularly in the behavioral and communicative

**Table 4. Maternal seroprevalence of CMV-IgG antibodies at birth and seroconversion comparing children with or without delayed language development at 3 years of age.**

| Maternal characteristics | Cases n = 299 | Controls n = 1349 |
|---|---|---|
| **CMV seroprevalence** | | |
| **CMV antibodies** | 189 | 754 |
| **Yes (IgG+)** | 63.2% | 55.9% |
| **CMV antibodies** | 110 | 595 |
| **No (IgG-)** | 36.8% | 44.1% |
| **CMV seroconversion [a]** | **Cases** | **Controls** |
| | **n = 118** | **n = 618** |
| **CMV seroconversion** | 8 | 23 |
| **Yes** | 6.8% | 3.7% |
| **CMV seroconversion** | 110 | 595 |
| **No** | 93.2% | 96.3% |

[a] Only seroconversion samples and seronegative (IgG-negative) in the M1 samples are taken into calculation (118 cases and 618 controls).

areas [23]. Both the asymptomatic and the symptomatic children received valganciclovir treatment, and the long-term sequelae was not affected by which of the three trimesters that the maternal infection occurred (21). Our results show a trend similar with these previous studies, although not statistical significant. The estimate of the association when hearing impairment was not adjusted for in the regression model was similar to the estimate when it was included. A similar result was also seen when we removed the hearing impaired children from the analysis. These results agree with the hypothesis that CMV could cause delayed language development independent of hearing impairment.

Seropositivity and seroconversion rates in our study are comparable to other studies among pregnant women. We found that variables, like parity and education level, influenced the CMV seropositivity and seroconversion rates [24, 25]. In contrast to other countries comparable to Norway, age did not influence the risk of being IgG positive in this and our previous study within MoBa [6]. This is consistent with observations in previous Norwegian studies and has been explained partly by the decades-long high percentage of breastfeeding women in

**Table 5. Results from logistic regression modelling, with crude and adjusted odds ratios for the association between CMV IgG+ and delayed language development.**

| | OR (95% CI) | P-value |
|---|---|---|
| **Model 1** | | |
| CMV IgG+ | 1.36 (1.05–1.76) | 0.021 |
| **Model 2** | | |
| CMV IgG+ | 1.29 (0.99–1.69) | 0.061 |
| Age | 1.02 (0.99–1.05) | 0.134 |
| Parity | 1.42 (1.08–1.88) | 0.013 |
| 12 + years of education | 0.39 (0.29–0.50) | <0.001 |
| **Model 3** | | |
| CMV IgG+ | 1.29 (0.99–1.71) | 0.064 |
| Hearing impairment | 4.26 (2.67–6.81) | <0.001 |
| Age | 1.03 (0.99–1.06) | 0.087 |
| Parity | 1.39 (1.05–0.86) | 0.022 |
| 12 + years of education | 0.39 (0.29–0.51) | <0.001 |

**Table 6. Results from logistic regression modelling, with crude and adjusted odds ratios for the association between CMV IgG seroconversion and delayed language development.**

| | OR (95% CI) | P-value |
|---|---|---|
| **Model 1** | | |
| CMV seroconversion | 1.88 (0.82–4.31) | 0.135 |
| **Model 2** | | |
| CMV seroconversion | 2.18 (0.92–5.17) | 0.076 |
| Age | 1.01 (0.96–1.06) | 0.61 |
| Parity | 1.28 (0.83–1.97) | 0.26 |
| 12 + years of education | 0.32 (0.21–0.49) | <0.001 |
| **Model 3** | | |
| CMV seroconversion | 1.91 (0.77–4.74) | 0.166 |
| Hearing impairment | 4.69 (2.30–9.59) | <0.001 |
| Age | 1.02 (0.97–1.07) | 0.51 |
| Parity | 1.26 (0.81–1.97) | 0.31 |
| 12 + years of education | 0.31 (0.20–0.48) | <0.001 |

Norway, and high attendance to day care centers for children where there is a high risk of CMV transmission [5, 26].

The strength of this study is the population-based cohort design. The cases and controls were drawn among more than 90 000 pregnant women participating in MoBa, a large prospective population based cohort over a 10- year time span. Another strength is that the participants had blood sampled at two occasions during their pregnancy, at week 18 and after birth, making it possible to test for seropositivity at two time points and checking for seroconversions. A limitation is the timing of inclusion of participants in the MoBa study. The participants were invited to MoBa during routine hospital examination with ultrasound in pregnancy weeks 17 or 18, when the first blood samples were taken. Thus, seroconversion occurring early in pregnancy or in the peri-conceptional period may not be detected, and incorrectly assumed to be a past CMV-infection. Another weakness is the exclusion of mothers of children selected as cases or controls in the ABC-study within MoBa, this may have influenced the eligible number of cases with delayed language development for this study [14]. This study was designed to only look at CMV in pregnancy as a cause of language development delay. Potential causes like chromosomal abnormalities was not investigated as this was an exclusion criteria in our study. Other disorders like inherited diseases or infections other than CMV were not investigated, and the study was not designed to have enough power to fully investigate the potential mediating effects of hearing impairment.

Investigation of CMV in newborns was not part of this study. Previously we have shown that cCMV is under-diagnosed and awareness on this matter should be strengthened [6]. There is lack of data on long-term outcome of newborns with cCMV infection particularly for those asymptomatic at birth. The influence of maternal CMV-infection, both primary and non-primary, upon language impairment should be further investigated, both in asymptomatic and symptomatic cCMV-children, especially without concomitant hearing impairment.

## Conclusion

This population-based study supported the previous study finding of a higher risk of delayed language development in children at three years of age in mothers seropositive for CMV than in mothers seronegative for CMV although no statistical significant association was found connected with CMV seroconversion.

## Acknowledgments

We are grateful to all the participating families in Norway who take part in The Norwegian Mother, Father and Child Cohort Study. We would like to thank Synnve Schjolberg for excellent advice regarding the language delay cases and, Hege Fremstad, and Moustafa Gibory, at the Norwegian Institute of Public Health for excellent technical assistance.

## Author Contributions

**Conceptualization:** Regine Barlinn, Susanne G. Dudman, Lill Trogstad, Per Magnus.

**Data curation:** Regine Barlinn, Halvor Rollag, Per Magnus.

**Formal analysis:** Regine Barlinn, Susanne G. Dudman, Lill Trogstad, Jonas C. Lindstrøm.

**Funding acquisition:** Susanne G. Dudman.

**Investigation:** Regine Barlinn, Susanne G. Dudman.

**Methodology:** Regine Barlinn, Susanne G. Dudman, Halvor Rollag, Lill Trogstad, Jonas C. Lindstrøm, Per Magnus.

**Project administration:** Susanne G. Dudman.

**Resources:** Regine Barlinn, Susanne G. Dudman.

**Software:** Regine Barlinn, Jonas C. Lindstrøm.

**Supervision:** Susanne G. Dudman, Halvor Rollag, Lill Trogstad, Per Magnus.

**Validation:** Regine Barlinn, Susanne G. Dudman, Halvor Rollag, Jonas C. Lindstrøm, Per Magnus.

**Writing – original draft:** Regine Barlinn, Susanne G. Dudman.

**Writing – review & editing:** Regine Barlinn, Susanne G. Dudman, Halvor Rollag, Lill Trogstad, Jonas C. Lindstrøm, Per Magnus.

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
