## [Decision Letter · Decision Letter 0]

9 Aug 2022

PONE-D-22-17172Maternal cytomegalovirus infection and delayed language development in children at 3 years of age – a nested case-control study in a large population-based pregnancy cohort

PLOS ONE

Dear Dr. Regine Barlinn, MD PhD,

Thank you for submitting your manuscript to PLOS ONE. After careful consideration, we feel that it has merit but does not fully meet PLOS ONE’s publication criteria as it currently stands. Therefore, we invite you to submit a revised version of the manuscript that addresses the points raised during the review process.

ACADEMIC EDITOR COMMENTS

The topic of the manuscript is interesting. Nevertheless, the reviewers raised several concerns: considering this point, I invite authors to perform the required major revisions.

A# Abstract

1. Introduction line 7-8, “Maternal CMV infection in pregnancy may lead to vertical transmission of CMV”. This sentence is incomplete and revised it.

The abbreviation is not recommended in abstract

2. Line 10-11…. with and without delayed language development in 3-year-old children. Why do you specifically take three years and below? It needs strong justification.

B. Methods

1. What are the abbreviations of MoBa?

The abbreviation is not recommended in abstract

3. Line 14-15 “Blood samples were obtained from mothers during pregnancy weeks 17 or 18 in pregnancy and after birth”. It needs clarification why you took blood at 17 or 18 weeks of gestations.

4. Elaborate on how to select control for your cohort study.

C. Results

1. Line 23-24 “the OR for seroconversion in cases as compared to control mothers was 1.88 (CI; 0.82 to 4.31). Seroconversion in cases is not associated with control mothers”. Please see again your analysis part.

D. Revised your conclusion based on your pertinent findings.

# Introduction

General comments

You put your introduction in two paragraphs

30-47 one paragraph and 48-51

Based on the above comments, you should revise your paragraph and split the ideas into different paragraphs.

* Introduction line 7-8, “Maternal CMV infection in pregnancy may lead to vertical transmission of CMV”. This sentence is incomplete and revised it.

* line 48 nested case-control study

# Methods

* Elaborate on how to calculate the sample size of cases and why you took 300 why not other numbers.

*Elaborate on how to calculate the sample size of controls and why you took 1350 why not other numbers.

*Line 82 you mean questionnaire 21 or where is the questionnaire?

Statistical analyses

1. Please elaborate “A sample of 300 cases and 1350 123 controls were needed to detect an odds ratio (OR) for CMV-infection of 1.5 with a significance level of 0.05 and power of 80%”. It is a sample size calculation?

2. How do control confounding and mediating variables? Elaborate it.

3. Which model do you use to cut off your points? Did you check your model’s fitness?

# Result

1. The maternal education level was lower in case mothers (49%) ≤12 years than in controls (27%).

2. It is not clear to me that≤12 years, you mean grade two?

3. Put confidence interval (OR > 4, p<0.001)

4. you should recategorize “maternal education level” again

#conclusion

Revised your conclusion based on your pertinent findings

you should write a declaration based on PLOS ONE guidelines,

Put all abbreviations

We look forward to receiving your revised manuscript.

Kind regards,

Zemenu Yohannes Kassa, Msc

Academic Editor

PLOS ONE

Journal Requirements:

5. Please ensure that you include a title page within your main document. You should list all authors and all affiliations as per our author instructions and clearly indicate the corresponding author. 

Reviewers' comments:

Reviewer's Responses to Questions

**Comments to the Author**

1. Is the manuscript technically sound, and do the data support the conclusions?

Reviewer #1: Partly

Reviewer #2: Yes

2. Has the statistical analysis been performed appropriately and rigorously? 

Reviewer #1: No

Reviewer #2: No

3. Have the authors made all data underlying the findings in their manuscript fully available?

Reviewer #1: No

Reviewer #2: Yes

4. Is the manuscript presented in an intelligible fashion and written in standard English?

Reviewer #1: Yes

Reviewer #2: Yes

5. Review Comments to the Author

Reviewer #1: I found this article very interesting aiming to describe the association between cCMV and delayed language development. However, there is main concerns in the statistical analysis.

The authors mentioned that CMV seropositive does not associate with hearing impairment. Then, hearing impairment is less likely a mediator between cCMV and delayed hearing development. if the authors think hearing impairment is the mediator variable, there are some other analysis methods, such as path analysis or mediator analysis.

Did authors try to conduct multi-variate logistic regression between delayed language development and seropositivity/seroconversion adjusting for hearing impairment? Is there any collinearity? Please clarify if those attempts were made.

I think hearing impairment will be a confounder when analyzing the association between cCMV infection and delayed language development if hearing impairment is not caused by cCMV only. Hearing impairment should be included in the model to adjust the effect rather than conducting sensitivity analysis.

The authors may misinterpret 95%CI of OR. There are several sentences that they have judged there is an association between outcome variable and explanatory variable, but 95%CI is crossing 1, which means there is no statistical significance in associations, e.g., lines 156 and 205.

I understand that maternal age, parity and education level were associated with CMV seropositivity and seroconversion (explanatory variables) but not all were associated with delayed language development (outcome variable), then they are not necessarily to be included in the model. On the other hand, BMI seems to have an association with seropositive, then why not including BMI? What is the variable selection strategy?

Table 3 should include crude ORs (and p-values) as a result of univariate logistic regression. CMV seropositive and negative should be in the row as one explanatory variable rather than separate column. Statistical analysis and Line 133-144 should be also rewritten according to the results in table 3.

Isn't it possible to include before 30 weeks of gestational age? I wonder why there is no children before 30 weeks. Or is there any cases with other known virus infection, such as rubella?

There are several minor comments:

line 36, is this seroprevalence maternal or women in reproductive age? please specify.

line 48-49, the association between CMV seropositivity and seroconversion in pregnant women with their children who had delayed language development. (delayed language development should be used consistently in the article as a outcome variable)

line 84 , please clarify the abbreviation of MBRN

Line 86 outcome measured. Is this different from case? this section should be combined with Case definition. (or deleted if duplicated.)

line 100, other variables

Is hearing impairment defined only by questionnaire? Any test result of any referral history to the otolaryngologist?

Can you briefly describe when BMI was measured and when GA, GW and other factors included in univariate analysis were collected?

line 121, hearing impairment was included in covariates to adjust the association, but the results Table 3,4,5 did not include this variable, rather the authors did separate analysis. It needs to be clarified.

In my understanding, only when M2 sample's IgG or IgM were positive, therefore M1 samples were tested, these samples were included in the analysis, correct? I think it is a bit unclear, so please clarify.

In discussion, it should be considered other explanations for delayed language development except cCMV. And if cCMV causes language development delay, what the potential mechanisms are. Even without any chromosomal abnormalities, are there any other development delay or growth retardation or other factors such as asphyxia at birth or inherited disorder and other reasons for admissions to the NICU? Whether children have siblings?

Reviewer #2: - The topic is so relevance and timely to improve maternal and child health

- The paper is well written , but need to address points raised in detail below

- Tables should be revised according to the comments below

- The discussion should address the limitation

- The result should be revised in brief description of the cohort outcome specifically to CMV incidence

My comments for authors in details are attached. Please forward the comments to authors to revise it.

6. PLOS authors have the option to publish the peer review history of their article (what does this mean?). If published, this will include your full peer review and any attached files.

Reviewer #1: No

Reviewer #2: **Yes: **Abel Gedefaw( MD, MPH), Associate Professor of Obstetrics and gynecology, Hawassa University , Ethiopia

---

## [Author Response · Author response to Decision Letter 0]

6 Sep 2022

Response to academic editor and reviewers

Ref. No.: PONE-D-22-17172

Title: Maternal cytomegalovirus infection and delayed language development in children at 3 years of age – a nested case-control study in a large population-based pregnancy cohort. 

First, I would like to thank the reviewers and the editor for the time and effort that has been put into assessing the previous version of the manuscript. Please find our point-by-point responses to the comments below.

ACADEMIC EDITOR COMMENTS

The topic of the manuscript is interesting. Nevertheless, the reviewers raised several concerns: considering this point, I invite authors to perform the required major revisions.

A# Abstract

1. Introduction line 7-8, “Maternal CMV infection in pregnancy may lead to vertical transmission of CMV”. This sentence is incomplete and revised it.

The abbreviation is not recommended in abstract. We have revised the introduction line 7-8 and included an explanation of the abbreviation. 

2. Line 10-11…. with and without delayed language development in 3-year-old children. Why do you specifically take three years and below? It needs strong justification.

The language development status was investigated at the age of three. Below this age the natural variation of language development is great making comparisons more difficult. A validation was conducted initially to assess the questionnaire’s ability to detect language development by comparing the parent`s response to concomitant psychological examination of the child at 3 years showing high correlation. This validation justified the use of questionnaire data collected at the age of 3 years.

B. Methods

1. What are the abbreviations of MoBa? The abbreviation is not recommended in abstract.

The abbreviation is explained in the article text and we have omitted the abbreviation from the abstract. The short name of the cohort study is MoBa as it is made up of the Norwegian words for mother (Mor) and child (Barn).

3. Line 14-15 “Blood samples were obtained from mothers during pregnancy weeks 17 or 18 in pregnancy and after birth”. It needs clarification why you took blood at 17 or 18 weeks of gestations.

As part of the pregnancy follow-up, all pregnant women in Norway are invited to ultrasound examination in week 17 to 18, when inclusion was done. It was convenient to draw the blood samples at this visit.

4. Elaborate on how to select control for your cohort study.

Using a statistical soft-ware (SPSS), the controls were randomly selected from the total cohort.

C. Results

1. Line 23-24 “the OR for seroconversion in cases as compared to control mothers was 1.88 (CI; 0.82 to 4.31). Seroconversion in cases is not associated with control mothers”. Please see again your analysis part.

We have revised the sentence.

D. Revised your conclusion based on your pertinent findings.

We have revised the conclusion in the manuscript and result section in the abstract.

# Introduction

General comments

You put your introduction in two paragraphs

30-47 one paragraph and 48-51

Based on the above comments, you should revise your paragraph and split the ideas into different paragraphs.

We kept the introduction in two paragraphs, since we believe it is easier for the reader.

* Introduction line 7-8, “Maternal CMV infection in pregnancy may lead to vertical transmission of CMV”. This sentence is incomplete and revised it.

We have revised the introduction line in abstract and introduction.

* line 48 nested case-control study

This has been revised accordingly.

# Methods

* Elaborate on how to calculate the sample size of cases and why you took 300 why not other numbers.

*Elaborate on how to calculate the sample size of controls and why you took 1350 why not other numbers.

A power calculation was performed initially in the planning of this study. The numbers of cases and controls needed to achieve a power of 80% and 95% confidence level were calculated based on 4% exposure to CMV infection among cases. The result of the power calculation showed that 300 cases and 1350 controls were needed using OpenEpi version 2013.

*Line 82 you mean questionnaire 21 or where is the questionnaire?

We are referring to MoBa questionnaire number 1 from pregnancy week 17-18 at inclusion of the study. Please see reference number 13. Question 21 was part of the 3 years of age questionnaire. We have revised the manuscript to clarify this matter.

Statistical analyses

1. Please elaborate “A sample of 300 cases and 1350 123 controls were needed to detect an odds ratio (OR) for CMV-infection of 1.5 with a significance level of 0.05 and power of 80%”. It is a sample size calculation? Yes, please see explanation above.

2. How do control confounding and mediating variables? Elaborate it. Please see explanation below.

3. Which model do you use to cut off your points? Did you check your model’s fitness? 

Cut off were based on questionnaire variables, and no model was used. The questionnaires have been previously validated for the purpose, please see reference 15.

# Result

1. The maternal education level was lower in case mothers (49%) ≤12 years than in controls (27%).

2. It is not clear to me that≤12 years, you mean grade two? 

4. you should recategorize “maternal education level” again

We have used dichotomous variable for education. The junior and high school last 12 years in Norway. We categorized the education level into ≤12 years and above 12 years of education.

3. Put confidence interval (OR > 4, p<0.001)

We have added the confidence interval as suggested.

#conclusion

Revised your conclusion based on your pertinent findings

you should write a declaration based on PLOS ONE guidelines,

Put all abbreviations

Thanks for these points, we have updated accordingly.

Reviewer #1: I found this article very interesting aiming to describe the association between cCMV and delayed language development. However, there is main concerns in the statistical analysis.

The authors mentioned that CMV seropositive does not associate with hearing impairment. Then, hearing impairment is less likely a mediator between cCMV and delayed hearing development. if the authors think hearing impairment is the mediator variable, there are some other analysis methods, such as path analysis or mediator analysis.

Did authors try to conduct multi-variate logistic regression between delayed language development and seropositivity/seroconversion adjusting for hearing impairment? Is there any collinearity? Please clarify if those attempts were made.

We have included results from the statistical analysis adjusted for hearing impairment. It is also important to note that the study was not designed nor powered to investigate any hypothetical mediation effects of hearing impairment, so we have not investigated this in any more depth. We have now mentioned this in the discussion as well.

I think hearing impairment will be a confounder when analyzing the association between cCMV infection and delayed language development if hearing impairment is not caused by cCMV only. Hearing impairment should be included in the model to adjust the effect rather than conducting sensitivity analysis.

We have added results of regression analyses both with and without hearing impairment as a predictor. Thus, we accommodate for hearing impairment being either mediator or confounder. If hearing impairment is a mediator, we also get what might be interpreted as a direct effect of CMV infection when hearing is adjusted for. 

The authors may misinterpret 95%CI of OR. There are several sentences that they have judged there is an association between outcome variable and explanatory variable, but 95%CI is crossing 1, which means there is no statistical significance in associations, e.g., lines 156 and 205.

We agree with the reviewer on the interpretation of 95% CI of OR. We have clarified in the discussion and results that the results are not statistically significant. 

I understand that maternal age, parity and education level were associated with CMV seropositivity and seroconversion (explanatory variables) but not all were associated with delayed language development (outcome variable), then they are not necessarily to be included in the model. On the other hand, BMI seems to have an association with seropositive, then why not including BMI? What is the variable selection strategy?

We believe that it is important to study potential association of age, parity and education level, therefore these variables were included in the statistical analyses. When looking at risk factors for maternal CMV infection, BMI is not likely to be of impact for acquiring CMV in contrast to age, parity and education level, which are factors previously found to be linked to higher rates of CMV seroprevalence. We judge that the observed association between BMI and CMV is confounded by the other variables.

Table 3 should include crude ORs (and p-values) as a result of univariate logistic regression. CMV seropositive and negative should be in the row as one explanatory variable rather than separate column. Statistical analysis and Line 133-144 should be also rewritten according to the results in table 3.

Table 3 has been changed accordingly. Also, a paragraph has been added in statistical analysis section and in the text.

Isn't it possible to include before 30 weeks of gestational age? I wonder why there is no children before 30 weeks. Or is there any cases with other known virus infection, such as rubella?

Women in the MoBa were invited to participate at week 17-18, and their pregnancy outcome were recorded for all. In our study, there were no exclusion criteria in case of premature birth. It is a coincidence that there were no premature birth prior week 30 among cases and controls. In Norway approximately 0.4 % of pregnancies result in birth between week 22-27, and 5% in birth between week 28-36 according to Medical Birth Registry of Norway.

As the objective of our study is cmv-infection, we did not investigate other virus infections such as rubella. However, it is unlikely that any rubella infection occurred among our study participants since rubella was eliminated many years ago from Norway (www.fhi.no). We have included a paragraph about this limitation in the discussion.

There are several minor comments:

line 36, is this seroprevalence maternal or women in reproductive age? please specify.

Seroprevalence reference is in women of reproductive age. We have added a new reference for the CMV seroprevalence among women of reproductive age. 

line 48-49, the association between CMV seropositivity and seroconversion in pregnant women with their children who had delayed language development. (delayed language development should be used consistently in the article as a outcome variable)

We agree with the reviewer about referring to delayed language development as an outcome variable and have reviewed the text to check that it is used consistently. 

line 84 , please clarify the abbreviation of MBRN

Medical Birth Registry of Norway. This has been explained in line 63 in the revised article.

Line 86 outcome measured. Is this different from case? this section should be combined with Case definition. (or deleted if duplicated.)

We have deleted duplicated text from this section.

line 100, other variables

Is hearing impairment defined only by questionnaire? Any test result of any referral history to the otolaryngologist?

Yes, it is correct that data on hearing impairment came from the questionnaire. Medical test results from otolaryngologist were not available.

Can you briefly describe when BMI was measured and when GA, GW and other factors included in univariate analysis were collected?

BMI was measured prior to the pregnancy through questionnaire 1. GA, GW and placenta weight data was collected from birth records in the MBRN. 

We have now added this information in footnote of table 3.

line 121, hearing impairment was included in covariates to adjust the association, but the results Table 3,4,5 did not include this variable, rather the authors did separate analysis. It needs to be clarified.

We have included hearing impairment in the statistical analysis as shown in the new tables 5 and 6. Also the text in the results and discussion section is revised accordingly.

In my understanding, only when M2 sample's IgG or IgM were positive, therefore M1 samples were tested, these samples were included in the analysis, correct? I think it is a bit unclear, so please clarify.

The laboratory analysis started with IgG and IgM in M2 sample. If there were any positive IgG or IgM results, the M1 sample was also analyzed. There were no need to analyze the M1 samples in case of negative IgG and IgM in M2 as explained in line 111-114.

In discussion, it should be considered other explanations for delayed language development except cCMV. And if cCMV causes language development delay, what the potential mechanisms are. Even without any chromosomal abnormalities, are there any other development delay or growth retardation or other factors such as asphyxia at birth or inherited disorder and other reasons for admissions to the NICU? Whether children have siblings?

Other explanations for delayed language development than cCMV can be other congenital diseases e.g. inherited disorders and injuries at birth. We have added a paragraph in the discussion explaining about this and that a limitation to our study is that we did not examine for all these explanations. Chromosomal abnormalities were part of the exclusion criteria for this study as stated in material and methods section. The factor whether the children have siblings is covered by the parity status of the cases and controls.

Reviewer #2: - The topic is so relevance and timely to improve maternal and child health

- The paper is well written , but need to address points raised in detail below

- Tables should be revised according to the comments below

- The discussion should address the limitation

- The result should be revised in brief description of the cohort outcome specifically to CMV incidence

My comments for authors in details are attached. Please forward the comments to authors to revise it.

General comment 

- The topic is so relevance and timely to improve maternal and child health 

- The paper is well written , but need to address points raised in detail below 

Abstract: 

 Conclusion: Only the odds of CMV seroprevalence positivity, which had a significant association, are included in the conclusion part. The non-significant association of seroconversion odds should also be addressed in the conclusion. 

We have included a sentence concerning this lack of association in the conclusion of the article.

Introduction: 

Only one study, a Dutch study, mentioned the effects of CMV infection and language development. It is good to include more studies on CMV infection and language development to justify the importance of the study. 

There are high numbers of articles investigating the connection between CMV and hearing impairment in contrast to the very few studies looking at delayed language development. We have cited five articles previously published with this subject as a main focus.

Other risk factors for delay language development like other infections could be mentioned briefly and the importance of CMV infection study can be elaborate. 

Other explanations have been added to the discussion.

Materials and methods

i. Study ( Cohort ) population description

1. Briefly describe the objective of the cohort for a better understanding of the paper 

The objective of the MoBa study has been added to this part of the article.

2. It is good to describe briefly what was done or measured for the cohort population related to CMV infection and outcome despite it was mentioned in other papers. The only measurement stated in the manuscript was blood sample was taken for seroconversion and 2nd trimester CMV positivity analysis. Despite we assumed nearly half of the primary infection end up with congenital infection based on previous data, it is good to include the following if it was measured.

- Diagnosis of congenital CMV infection 

- Diagnosis of symptomatic CMV infection at birth 

- Whether there were follow-ups of infants born with CMV converted mothers for short and long-term sequel 

Examination of cCMV at birth and follow-ups of infants was not part of this study. Standard operating procedures on the diagnosis of cCMV and CMV with symptoms at birth has been briefly explained below. We have added some lines in the discussion about cCMV.

ii. Selection of cases need clarification. 

It is stated only as 300 cases of delayed language development. Among the 95,200 pregnant women cohort 

- How many children did have delayed language development 

- How did you select the 300 cases? How many cases were excluded from analysis due to other exclusion criteria ( Multiple pregnancies, chromosomal disorders, other study participation ) 

Please see explanation above.

Other variables: 

Only educational status mentioned. But in the result part, age, parity, gestational age, BMI, mentioned. 

We have added explanations of the following variables: age, parity, gestational age, BMI in table 3.

Statistical analysis 

 It is good to add the reference for sample size calculation 

Please see explanation above.

Result:

Despite it is case –control study, it is good to include incidence data related to CMV infection among the cohort population since it is a national and large sample cohort 

- What is the incidence of CMV positivity at the 2nd trimester (could be primary infection in the first trimester or reactivation of latent infection) and seroconversion rate at birth? 

We have previously studied the incidence of CMV positivity at 2nd trimester and found 54.1% seropositivity (Barlinn et al. APMIS 2018). We further investigated seroconversion rate in the cohort between week 17/18 and birth which was 3.7%. We have added this information in the text and included this study as a reference.

- Among the total seroconverted women 

 How many of them had been diagnosed with congenital CMV infection? 

 How many of them were symptomatic at birth for CMV infections? 

If there was no congenital infection diagnosis done, it should be stated in the discussion part regard to the clinical benefit of only maternal seroconversion at birth without a diagnosis of congenital infection. If we decrease the cases by half considering only half of the primary infection will have a congenital infection, do the risk of delayed language development association still persist

Investigation of cCMV at birth was not part of this study and this is now stated in the discussion. All new-borns in Norway are examined for the symptoms of cCMV and CMV PCR will be performed in saliva, urine and blood samples for diagnosis of CMV infection if this is suspected clinically. Additionally, all new-borns are subjected to hearing test and those failing this test will be without delay investigated for CMV by PCR in the same types of specimens as stated above.

- Table 3: add the P-Value for each variable to know the presence of association 

- Table 4: it is good to show the confounding variables crud and adjusted odd ratio too( parity, age, education, hearing impairment, etc for delayed language development 

This is now included in the new tables 5 and 6.

Discussion 

It states “ Seropositivity and seroconversion rates in our study are comparable to other studies among pregnant women”. This is difficult to conclude in case-control study. As mentioned above, we need to know the cohort result. 

It is good to discuss the limitation of 

1. Do we exclude other congenital infections such as syphilis, toxoplasmosis, and rubella based on maternal and infants’ serological testing? How do we control the confounding effect of other infections? Despite it is a case-control study and we need to mention the importance of excluding other infections in the limitation part. 

Other causes of congenital infections have not been investigated in our study, this has now been added to the limitations.

2. Ascertainment of congenital infection ( Seroconversion Vs congenital infection and language development delay ) 

Please see explanation above.

Conclusion: see above in the abstract

---

## [Decision Letter · Decision Letter 1]

26 Sep 2022

PONE-D-22-17172R1Maternal cytomegalovirus infection and delayed language development in children at 3 years of age – a nested case-control study in a large population-based pregnancy cohortPLOS ONE

Dear Dr. Regine Barlinn, MD PhD,,

Thank you for submitting your manuscript to PLOS ONE. After careful consideration, we feel that it has merit but does not fully meet PLOS ONE’s publication criteria as it currently stands. Therefore, we invite you to submit a revised version of the manuscript that addresses the points raised during the review process.

We look forward to receiving your revised manuscript.

Kind regards,

Zemenu Yohannes Kassa, Msc

Academic Editor

PLOS ONE

Additional Editor Comments:

Dear Dr. Regine Barlinn, MD PhD,

Academic editors’ comments

The topic of the manuscript is interesting and your manuscript is improved. However, the reviewers have raised concerns your statistical analysis how to control a confounder, you should check Multicollinearity and multivariable analysis. if you have a mediator variables, you should use other methods of analysis: considering this point, I invite authors to perform the required major revisions.

Reviewers' comments:

Reviewer's Responses to Questions

**Comments to the Author**

1. If the authors have adequately addressed your comments raised in a previous round of review and you feel that this manuscript is now acceptable for publication, you may indicate that here to bypass the “Comments to the Author” section, enter your conflict of interest statement in the “Confidential to Editor” section, and submit your "Accept" recommendation.

Reviewer #1: (No Response)

Reviewer #2: (No Response)

2. Is the manuscript technically sound, and do the data support the conclusions?

Reviewer #1: Partly

Reviewer #2: Yes

3. Has the statistical analysis been performed appropriately and rigorously? 

Reviewer #1: No

Reviewer #2: Yes

4. Have the authors made all data underlying the findings in their manuscript fully available?

Reviewer #1: Yes

Reviewer #2: (No Response)

5. Is the manuscript presented in an intelligible fashion and written in standard English?

Reviewer #1: Yes

Reviewer #2: Yes

6. Review Comments to the Author

Reviewer #1: The manuscript became clearer and improved.

I recognized that the authors main interest is the association between maternal CMV infection defined as serological positive or seroconversion and delayed language development regardless of the co-existence of hearing impairment.

The outcome is defined as moderate or severe problem of in ASQ question 18 (speech delay) AND simultaneous score <=40 in ASQ question 21 (cognitive developmental delay).

Authors answered all the questions, but I would like to clarify the major issue on the statistical analysis again.

1. If the authors think that hearing impairment is a mediator, they should not include the hearing impairment in the multivariate logistic regression in (Table 5 and 6). Just use model 1 and model 2.

2. If the authors think that hearing impairment is a confounder, they can keep the hearing impairment in the multivariate logistic regression and probably model 2 is just CMV IgG+ or seroconversion and hearing impairment, which is the authors' main interest. It is up to them to include other variables in model 3.

So please change the wording from mediator to confounder if authors would like to keep the current analysis. (line 198)

Alsos no need to mention mediator analysis in lines 269-271 if authors are going to take hearing impairment as a confounder.

I have one question in the definition of the case. ASQ question 21 is detecting the mental delay. I think the authors would like to remove this question 21 from the case definition as the language development delay without cognitive disorder was excluded in the current definition.

I also suggest that column of case and controls in table3 and table 4 should be combined as a result of univariate logistic regression analysis. CMV 1gG seropositive and seronegative should be in an independent table before Table 3.

Table 3 should include the age as a potential confounder.

In statistical analysis lines 147-149, it should be "Odds ratios with 95% confidence intervals were calculated with univariate logistic regression models, and adjusted ORs were calculated multivariate logistic regression adjusting for hearing impairment (if hearing impairment is recognized as a confounder), age, parity and maternal educational status." IF sensitivity analysis is going to be conducted, please mention that in the statistic section.

Line 250 authors said that age did not influence the risk of being IgG positive in our study. This result was not mentioned in the result section. I suggest including the age in table 3 or new table.

Conclusion said that "this population based study showed a higher risk of delayed language development ..." But I suggest "showed" is a bit strong conclusion while this study did not show the association between them. The expression may be,, "this study supported the previous study finding of ......."

Reviewer #2: (No Response)

7. PLOS authors have the option to publish the peer review history of their article (what does this mean?). If published, this will include your full peer review and any attached files.

Reviewer #1: No

Reviewer #2: **Yes: **Abel Gedefaw ( MD,MPH), Associate Professor of Obstetrics and Genecology, Hawassa University Collage of Medicine and Health Sciences , Hawassa , Ethiopia.

---

## [Author Response · Author response to Decision Letter 1]

19 Oct 2022

Reviewer response letter

Ref. No.: PONE-D-22-17172R1

Title: Maternal cytomegalovirus infection and delayed language development in children at 3 years of age – a nested case-control study in a large population-based pregnancy cohort. 

We would again like to thank the reviewers and the editor for the time and effort that has been put into assessing the previous version of the manuscript. Please find our point-by-point responses to the comments below.

Review Comments to the Author

Reviewer #1: The manuscript became clearer and improved.

I recognized that the authors main interest is the association between maternal CMV infection defined as serological positive or seroconversion and delayed language development regardless of the co-existence of hearing impairment.

The outcome is defined as moderate or severe problem of in ASQ question 18 (speech delay) AND simultaneous score <=40 in ASQ question 21 (cognitive developmental delay).

Please see explanation below about question 18 and 21.

Authors answered all the questions, but I would like to clarify the major issue on the statistical analysis again.

1. If the authors think that hearing impairment is a mediator, they should not include the hearing impairment in the multivariate logistic regression in (Table 5 and 6). Just use model 1 and model 2.

2. If the authors think that hearing impairment is a confounder, they can keep the hearing impairment in the multivariate logistic regression and probably model 2 is just CMV IgG+ or seroconversion and hearing impairment, which is the authors' main interest. It is up to them to include other variables in model 3.

So please change the wording from mediator to confounder if authors would like to keep the current analysis. (line 198)

Alsos no need to mention mediator analysis in lines 269-271 if authors are going to take hearing impairment as a confounder.

Response to poins 1 and 2:

The reviewer is correct that our main interest of the study is to ascertain the effect of CMV on speech development, but because the role of hearing impairment is unclear we have tried to account for this in our analyses and how we present our results. We do not think it is wise to commit to one of the two possibilities and pretend the other is irrelevant. We would also like to point out that if hearing impairment is indeed a mediator, it still makes sense to include an analysis where hearing impairment is a predictor. In the mediator case this would make the coefficients for CMV interpretable as direct effects. Hence, the model with hearing impairment included is relevant for both the mediator and confounder scenarios. 

We revised Results and Discussion to clarify this issue. Please see lines 153-155, 185, 205 and 251 – 254.

I have one question in the definition of the case. ASQ question 21 is detecting the mental delay. I think the authors would like to remove this question 21 from the case definition as the language development delay without cognitive disorder was excluded in the current definition.

The two questions are both exploring the language development in children. They were not developed to detect mental delay. Question 18 describes the speech performance and question 21 the child’s ability regarding language comprehension. The two questions are used in combination to select cases in the most appropriate way to ensure a consolidated group of children with language problems. A validation was conducted initially to assess the questionnaire’s ability to detect language development by comparing the parent`s response to these questions to a concomitant psychological examination of the child at 3 years showing high correlation. We have included a paragraph in the re-revised manuscript about this comparison and a new reference.

The selection of cases in this study is based on both Question 18 and 21 to obtain a consolidated group of children with language problems. This means that both questions are required for the analyses and Question 21 cannot be omitted from this study. 

I also suggest that column of case and controls in table3 and table 4 should be combined as a result of univariate logistic regression analysis. CMV 1gG seropositive and seronegative should be in an independent table before Table 3.

Table 3 should include the age as a potential confounder.

In the revised Table 3 we have included a column with p-values for comparing all the variables between cases and controls. We have included a new footnote with an explanation to this variable. This table also compares mean age between cases and controls as well as between CMV seropositives and seronegatives. No difference was found comparing age between the groups. 

Results section also includes Table 4 with a comparison of CMV IgG antibody status between cases and controls.

In statistical analysis lines 147-149, it should be "Odds ratios with 95% confidence intervals were calculated with univariate logistic regression models, and adjusted ORs were calculated multivariate logistic regression adjusting for hearing impairment (if hearing impairment is recognized as a confounder), age, parity and maternal educational status." IF sensitivity analysis is going to be conducted, please mention that in the statistic section.

We have revised accordingly in the Statistical analyses.

Line 250 authors said that age did not influence the risk of being IgG positive in our study. This result was not mentioned in the result section. I suggest including the age in table 3 or new table.

We thank the reviewer for the constructed comment and we understand that the sentence could be misunderstood. We have revised this sentence and included the reference to the previous study.

It is also referred to age in the beginning of the result section, stating that age was similar for CMV seropositive status compared to seronegative. Table 3, already includes age as a variable.

Conclusion said that "this population based study showed a higher risk of delayed language development ..." But I suggest "showed" is a bit strong conclusion while this study did not show the association between them. The expression may be,, "this study supported the previous study finding of ......."

We thank the reviewer for suggesting a revised Conclusion, which we have implemented.

---

## [Decision Letter · Decision Letter 2]

21 Nov 2022

Maternal cytomegalovirus infection and delayed language development in children at 3 years of age – a nested case-control study in a large population-based pregnancy cohort

PONE-D-22-17172R2

Dear Dr. Regine Barlinn,

We’re pleased to inform you that your manuscript has been judged scientifically suitable for publication and will be formally accepted for publication once it meets all outstanding technical requirements.

Kind regards,

Zemenu Yohannes Kassa, Msc

Academic Editor

PLOS ONE

Additional Editor Comments (optional):

Reviewers' comments:

Reviewer's Responses to Questions

**Comments to the Author**

1. If the authors have adequately addressed your comments raised in a previous round of review and you feel that this manuscript is now acceptable for publication, you may indicate that here to bypass the “Comments to the Author” section, enter your conflict of interest statement in the “Confidential to Editor” section, and submit your "Accept" recommendation.

Reviewer #1: (No Response)

Reviewer #3: (No Response)

2. Is the manuscript technically sound, and do the data support the conclusions?

Reviewer #1: Partly

Reviewer #3: Yes

3. Has the statistical analysis been performed appropriately and rigorously? 

Reviewer #1: (No Response)

Reviewer #3: Yes

4. Have the authors made all data underlying the findings in their manuscript fully available?

Reviewer #1: No

Reviewer #3: No

5. Is the manuscript presented in an intelligible fashion and written in standard English?

Reviewer #1: Yes

Reviewer #3: Yes

6. Review Comments to the Author

Reviewer #1: I am very sorry that my intention was not clearly understood by the authors.

I do understand that hearing impairment could be a mediator in this study design, but I cannot accept that a mediator is included as a covariate in a multivariate logistic regression model. I do not have adequate statistical knowledge how to handle this case if authors do not take hearing impairment as a confounder. At this point, I need to reject this manuscript, but if it is a too strict decision, I would like to ask editors or another reviewer who is more familiar with the cCMV and statistical analysis to make an appropriate decision.

Reviewer #3: A research study was conducted which aimed to estimate the odds ratios for CMV seropositivity and seroconversion in mothers, with and without delayed language development in 3 year old children. The observed odds ratio was 1.36 and the 95% CI did not contain zero, indicating a higher risk of delayed language development at three years of age in children whose mothers were seropositive for CMV, compared to children of mothers who were seronegative.

Minor revisions

Table 3:

A. In the statistical analysis section, state and describe the statistical methods used to estimate the p-values.

B. For continuous factors, provide standard deviations that correspond to means.

7. PLOS authors have the option to publish the peer review history of their article (what does this mean?). If published, this will include your full peer review and any attached files.

Reviewer #1: No

Reviewer #3: No

---

## [Editor Report · Acceptance letter]

23 Nov 2022

PONE-D-22-17172R2 

Maternal cytomegalovirus infection and delayed language development in children at 3 years of age – a nested case-control study in a large population-based pregnancy cohort 

Dear Dr. Barlinn:

I'm pleased to inform you that your manuscript has been deemed suitable for publication in PLOS ONE. Congratulations! Your manuscript is now with our production department. 

Kind regards, 

on behalf of

Dr. Zemenu Yohannes Kassa 

Academic Editor

PLOS ONE